# Effect of Digital Technologies on the Marginal Accuracy of Conventional and Cantilever Co–Cr Posterior-Fixed Partial Dentures Frameworks

Celia Tobar, Verónica Rodríguez, Carlos Lopez-Suarez, Jesús Peláez * , Jorge Cortés-Bretón Brinckmann and María J. Suárez

Department of Conservative Dentristy and Bucofacial Prosthesis, Faculty of Odontology, University Complutense of Madrid, 28040 Madrid, Spain; cetobar@ucm.es (C.T.); veranicr@ucm.es (V.R.); carlop04@ucm.es (C.L.-S.); jcortesb@ucm.es (J.C.-B.B.); mjsuarez@ucm.es (M.J.S.)
* Correspondence: jpelaezr@ucm.es

**Abstract:** The introduction of new digital technologies represents an important advance to fabricate metal–ceramic restorations. However, few studies have evaluated the influence of these technologies on the fit of the restorations. The aim of this study was to evaluate the effect of different manufacturing techniques and pontic design on the vertical marginal fit of cobalt—chromium (Co–Cr) posterior fixed partial dentures (FPDs) frameworks. Methods: Eighty stainless-steel dies were prepared to receive 3-unit FPDs frameworks with intermediate pontic (n = 40) and cantilever pontic (n = 40). Within each design, the specimens were randomly divided into four groups (n = 10 each) depending on the manufacturing technique: casting (CM), direct metal laser sintering (LS), soft metal milling (SM), and hard metal milling (HM). The frameworks were luted, and the vertical marginal discrepancy was assessed. Data analysis was made using Kruskal–Wallis and Mann–Whitney U tests ($\alpha = 0.05$). Results: The vertical marginal discrepancy values of all FPDs were below 50 μm. The HM frameworks obtained the lowest misfit values in both designs. However, no differences were found among intermediate pontic groups or cantilevered groups. Likewise, when differences in a marginal discrepancy between both framework designs were analyzed, no differences were observed. Conclusions: The analyzed digital technologies demonstrated high precision of fit on Co–Cr frameworks and on both pontic designs.

**Keywords:** marginal adaptation; fixed partial denture; dental technology; cobalt–chromium alloys; scanning electron microscopy

## 1. Introduction

Metal–ceramic restorations are still the most widely used for fixed prosthodontics due to their reliability and good long-term prognosis, widely tested, which has led that they are considered to be the gold standard [1–4]. Base metal alloys, especially cobalt–chromium (Co–Cr) alloy, have undergone a higher development in recent decades as an alternative to the costly noble alloys and the lower biocompatibility of nickel-chromium alloys [5,6]. In addition to good biocompatibility, Co–Cr alloys show proper corrosion stability and appropriate mechanical properties, as fracture resistance, hardness and resilience [7–10]. Metal–ceramic restorations have been processed by traditional casting techniques, but currently, new prosthetic technologies have been introduced to fabricate metal-base restorations. Nowadays, Co–Cr frameworks can be processed by computer-aided design and computer-aided manufacturing (CAD-CAM), involving subtractive and additive manufacturing processes. The subtractive or milling method consists of a process controlled by a computer program, which uses power-driven machine tools with a sharp cutting tool to mechanically cut the materials and to achieve specific geometries [11,12]. The main advantages are time-saving, the ability to create fine and precise detail, and the availability

of materials [11,13–15]. Nevertheless, it has disadvantages like the high cost and the waste of material [11,16]. Nowadays, metal milling frameworks can be obtained by hard or soft blocks. Soft metal milling is the most recent process within the subtractive method and consists of using metal blanks in a pre-sintered state. These blanks are milled and subsequently sintered in a special furnace until achieving the proper size (volumetric shrinkage of approximately 11%) [13]. This method has the advantages of shorter milling time, fewer machine tools wear (increasing its useful life), and less risk of material contamination due to dry milling [13]. On the other hand, the additive method, especially the direct metal laser sintering (DMLS), produces the metal structures layer by layer by a high-power laser that fuses the alloy powder from a three dimensional (3D) CAD file that contains the framework's design [17–19]. The advantages include no material waste, higher productivity [11], and easy fabrication of complex shapes [18,20]. The main drawbacks are that there may be differences in the final model production and limitations on materials so far [11,12].

A good marginal fit is one of the main criteria to achieve long-term success in fixed prostheses restorations. The lack of an accurate fit can cause severe complications [21–23], and multiple factors, such as tooth preparation, luting procedure, prosthetic design and manufacturing technology, can affect the final adaptation of the restorations [24–28].

Nevertheless, despite its importance, no consensus exists in the literature on what must be considered the optimal fit value [28–32], and most authors continue to refer to the criteria established by McLean and von Fraunhofer [33], which established as clinically acceptable a marginal discrepancy lower than 120 µm. Currently, CAD-CAM restorations have shown high precision in their marginal adaptation, and several authors admit gaps below 100 µm [28,32,34–38]. Different methods have been proposed to measure the marginal adaptation of a restoration. Direct-view microscopic techniques are the most commonly used, although there is no consensus in the methodology and the best technique to follow [26,27,39]. Scanning electron microscopy (SEM) is a conservative method to provide appropriate and realistic marginal fit observations with high magnification, especially with complex margin morphologies [27,28,39]. Nonetheless, this method also has disadvantages as the location of reference points for measurements and the observation angle [26–28,39,40].

There is limited information available regarding the influence of manufacturing techniques on the marginal adaptation of Co–Cr frameworks; thus, it is important to investigate the precision of fit of these restorations. Therefore, the present in vitro study's purpose was to evaluate and compare the vertical marginal discrepancy of posterior Co–Cr fixed partial dentures (FPDs) fabricated with different technologies and with two types of framework design (intermediate or cantilever pontic). The null hypotheses to be tested were that no differences in marginal fit would be found among the Co–Cr frameworks fabricated by the different technologies and framework designs.

## 2. Materials and Methods

### 2.1. Experimental Model

Eighty standardized machined stainless-steel master dies, with two abutments and a platform, were fabricated (Mechanical Workshop of Physical Science, University Complutense of Madrid, Spain). The platforms (30 mm in length, 17 mm in width, and 4.5 mm in thickness) [29,39] were designed to receive two designs of posterior 3-unit frameworks: (1) with an intermediate pontic (5 mm between abutments) (n = 40), and (2) cantilever pontic (0.2 mm between abutments) (n = 40). The abutments (n = 160) were designed simulating a first mandibular premolar prepared (5 mm in height, occlusal diameter of 5 mm, a 1 mm-wide chamfer circumferentially finish line, and a 6° angle of convergence of the axial walls) [28,29,32,34,36,39], and randomly screwed on the platform (Figures 1 and 2).

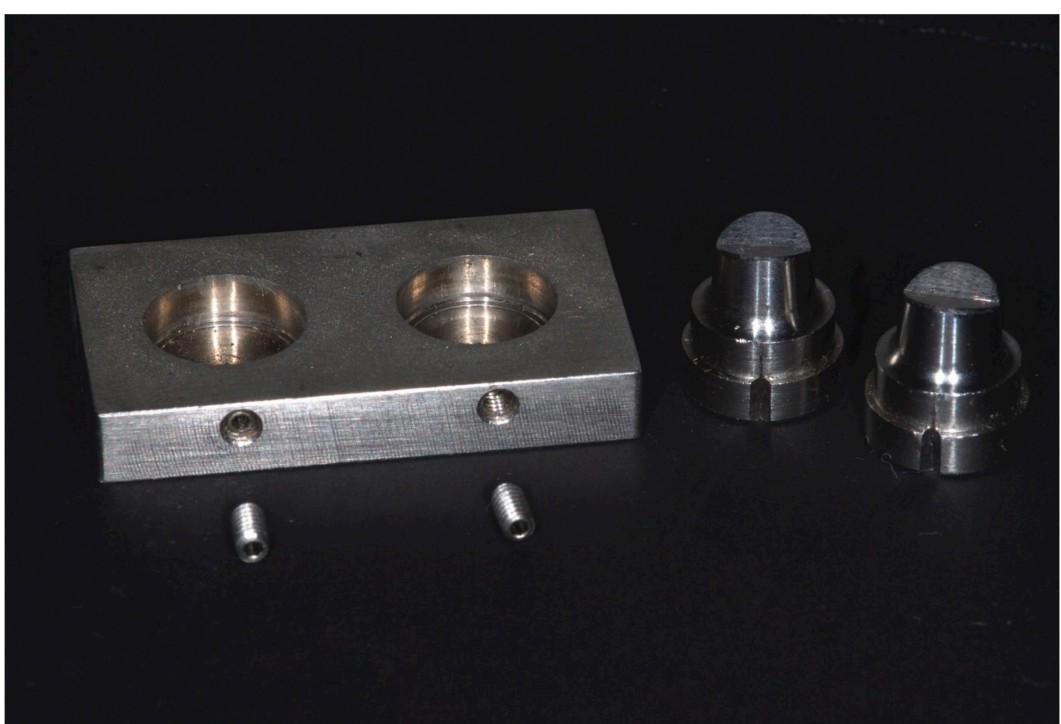

**Figure 1.** Master die components (platform, screws and abutments) of conventional frameworks.

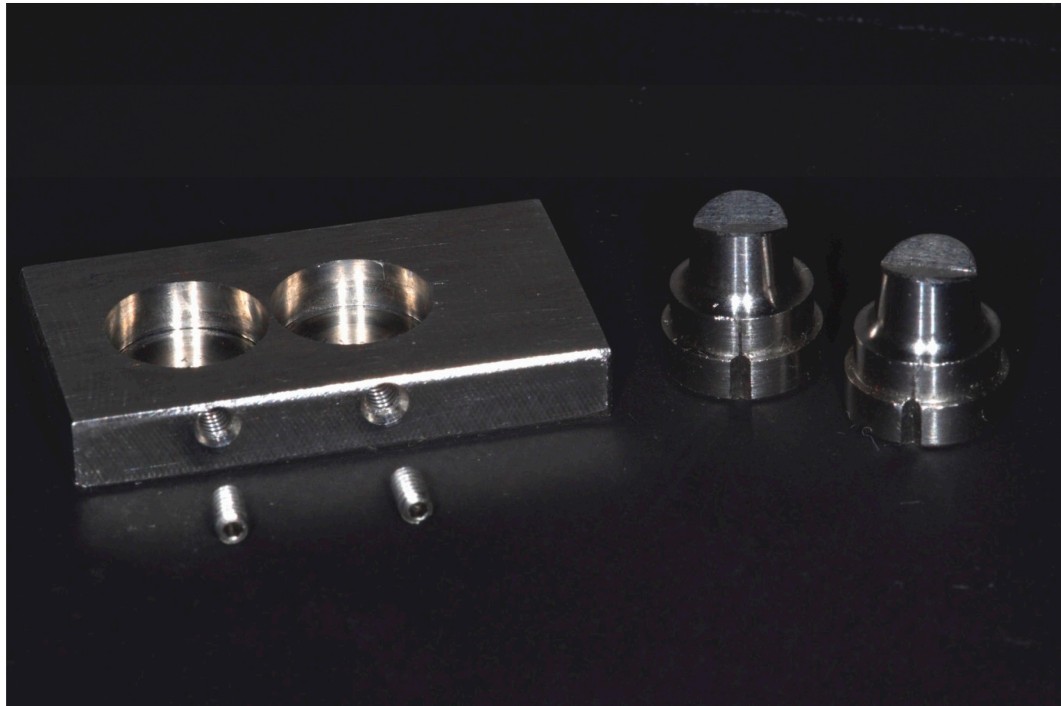

**Figure 2.** Master die components (platform, screws and abutments) of frameworks with cantilever pontic.

Within each design group, the specimens were randomly divided into four subgroups (n = 10 each, in accordance with the results of power analysis) depending on the manufacturing technique used to fabricate the frameworks: casting (CM), laser sintering (LS), soft metal milling (SM) and hard metal milling (HM). The specimens were used as working dies.

Table 1 displays the group code, coping alloys brands, composition, and manufacturers used in the study.

**Table 1.** Manufacturing technique, brands, manufacturers and chemical composition of the alloys selected for the study (weight %).

| Manufacturing Technique | | | | | | | | | | | | |
|---|---|---|---|---|---|---|---|---|---|---|---|---|
| **Group Code** | | Coping Alloys Brands and Manufacturers | **Dental Alloy Composition (Weight %)** | | | | | | | | | |
| **(1) Intermediate Pontic** | **(2) Cantilever Pontic** | | **Co** | **Cr** | **Mo** | **W** | **Si** | **Fe** | **C** | **Mn** | **Ni** | **N** |
| CM | CMc | Super 8 (Dental Alloys Products, San Diego, CA, USA) | 59.5 | 31.5 | 5 | - | 2 | ≤1 | ≤1 | ≤1 | - | - |
| LS | LSc | ST2724G (Sint-Tech, Clermont-Ferrand, France) | 65 | 28–30 | 5–6 | - | ≤1 | ≤0.5 | ≤0.02 | ≤1 | ≤1 | - |
| SM | SMc | Ceramill® Sintron R 71 L (Amann Girrbach, Koblach, Austria) | 66 | 28 | 5 | - | ≤1 | ≤1 | ≤0.1 | ≤1 | - | - |
| HM | HMc | Starbond CoS DISC basic (Scheftner, Mainz, Germany) | 59 | 25 | 3.5 | 9.5 | 1 | ≤1.5 | ≤1.5 | ≤1.5 | - | ≤1.5 |

### 2.2. Fabrication of the Restorations

To fabricate the CM and CMc frameworks, the specimens were scanned and digitized with the Lava Scan ST (3M ESPE, Seefeld, Germany), and the frameworks were designed using CAD software (DWOS version 7.0; Dental Wings, Montreal, QC, Canada). The wax patterns were made with the ProJet 1200 3D printer (3D Systems, Rock Hill, SC, USA) and invested with phosphate graphite-free investment plaster (Vestofix; DFS Diamond GmbH, Riedenburg, Germany). The casting was performed using induction and a centrifugal vacuum-casting machine (MIE-200C/R; Ordenta, Arganda del Rey, Spain) under vacuum pressure of 580 mmHg, at a melting temperature of 1480 °C. After casting, the samples were cleaned with water steam and sandblasted with aluminum-oxide particles (50 μm) under 50 N/cm$^2$ pressure (EXTRAmatic 9040; Kavo Dental GmbH, Biberach, Germany).

To prepare the LS and LSc frameworks, the scanning and design process was similar to the CM and CMc groups. The CAD design file was transferred to a DMLS unit (PM 100 Dental; Phenix Systems, Clermont-Ferrand, France), and the laser sintering process was performed by building 20 mm layers of alloy powders from the occlusal surface to the margins by applying a Yb-fiber laser at 1650 °C under an argon atmosphere. All the frameworks were cleaned and sandblasted in the same manner as the casted frameworks

To fabricate the SM and SMc frameworks, the specimens were scanned (Ceramill Map400; Amann Girrbach, Koblach, Austria), and the data were entered into specific design software (Ceramill Mind; Amann Girrbach). To compensate for the post-sintering shrinkage, the design was enlarged by 11%. These data were pre-set in the software. The frameworks were manufactured from pre-sintered Co–Cr discs in a milling unit (Ceramill Motion 2; Amann Girrbach). Then, the specimens were placed in a sintering tray (Ceramill Argovent; Amann Girrbach) and introduced into a sintering furnace (Ceramill Algotherm 2; Amann Girrbach) at 1.300 °C under an argon atmosphere to prevent oxidation. All the frameworks were cleaned and sandblasted in the same manner as the casted frameworks.

The manufacturing process for the HM and HMc frameworks also began with scanning the specimens (3Shape D750; 3Shape Dental System, Copenhagen, Denmark) and designing the copings by the specific software (Molder Builder; 3Shape Dental System). Two sintered Co–Cr discs were inserted in the warehouse (PH 2/120 SAUER; DGM Mori, Stipshausen, Germany) of the milling unit (Ultrasonic 10 linear; DMG Mori, Bielefeld, Germany) and machining was carried out.

All the 3D framework designs were done by experienced technicians with the same parameters: 0.5 mm wall-thickness, internal cement space of 50 μm, a premolar shape pontic, and a connector area of 9 mm$^2$ (3 mm × 3 mm). All the frameworks were cleaned and sandblasted in the same manner as the casted frameworks

The frameworks were luted onto their corresponding specimen using conventional glass–ionomer cement (Ketac-Cem EasyMix; 3 M ESPE), mixed following the manufacturer's instructions, at room temperature (18–24 °C) and relative humidity (50 ± 10%). The

cement was placed on the axial walls of the structures, and a constant seating force of 50 N was applied with a torque wrench (Ziacom, Madrid, Spain) fitted to a customized device (Mechanical Workshop of Physical Science, University Complutense of Madrid, Spain) for 10 min.

The marginal accuracy of the restorations was measured under a SEM (JSM-6400; JEOL, Tokyo, Japan) to determinate the vertical marginal discrepancy (or the vertical distance between the restoration margin and the preparation cavosurface angle, measured parallel to the longitudinal axis of the tooth [41]) (Figure 3). The specimens were coated with 24 kt, 19.32 g/m³ density gold by a Q150RS metallizer (Quorum Technologies, Laughton, United Kingdom) before SEM evaluation and then positioned in a customized clamp perpendicular to the axis of the microscope [32]. To standardize the marginal evaluation, the measuring areas were marked at the same point in the middle of the buccal and lingual surfaces of each abutment in the gap region with an indelible marking pen (Lumocolor permanent; Staedler Mars, Nuremberg, Germany) [28,32,34,39]. The SEM was connected to a computer with the INCA Suite version 4.04 software (Oxford Instruments; Abingdon, Oxfordshire, UK), which was run to capture and calibrate the images at the marked areas, at ×1000 magnification (Figure 4). All the images were captured by the same operator with an acceleration voltage of 20 KV and a 20 mm working distance, achieving a eucentric position, which means that the position at which the primary electron beam hits does not change from one sample to another by following the same coordinates [42]. To increase the number of measurements per specimen, the images were edited by using imaging software (ImageJ version 1.49; U.S. National Institutes of Health, Bethesda, MD, USA), creating lines parallel to the original. Therefore, 60 measurements were recorded for each specimen (30 per abutment) (Figure 5).

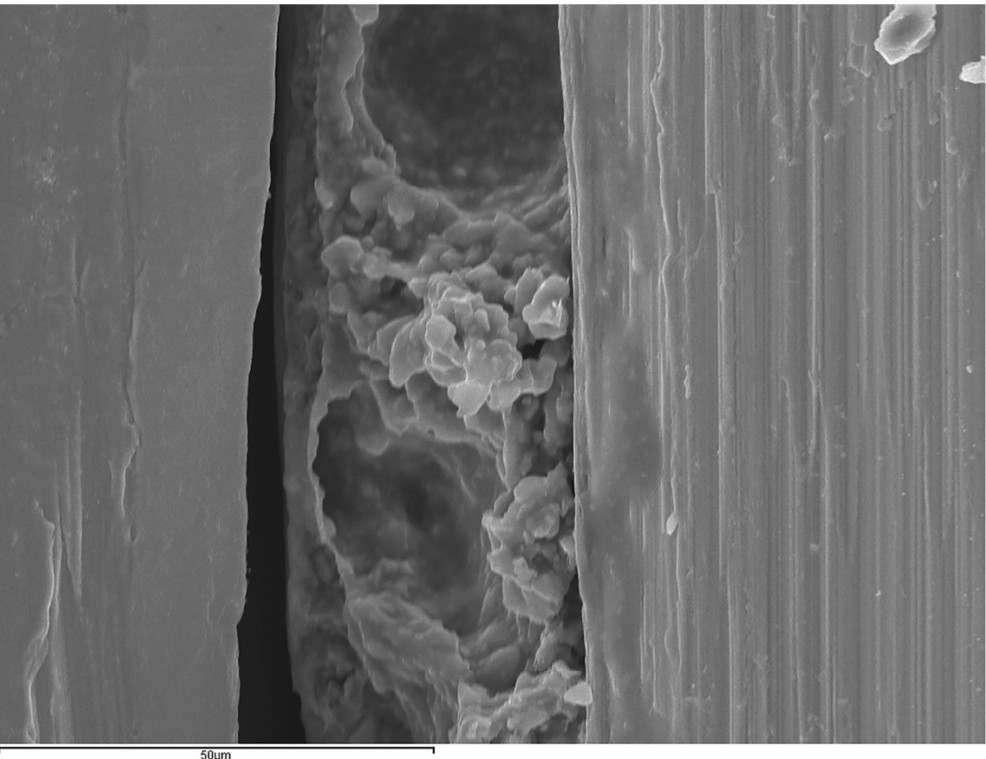

**Figure 3.** SEM image (1000×) showing the marginal gap in a representative framework of the hard metal milling (HM) group.

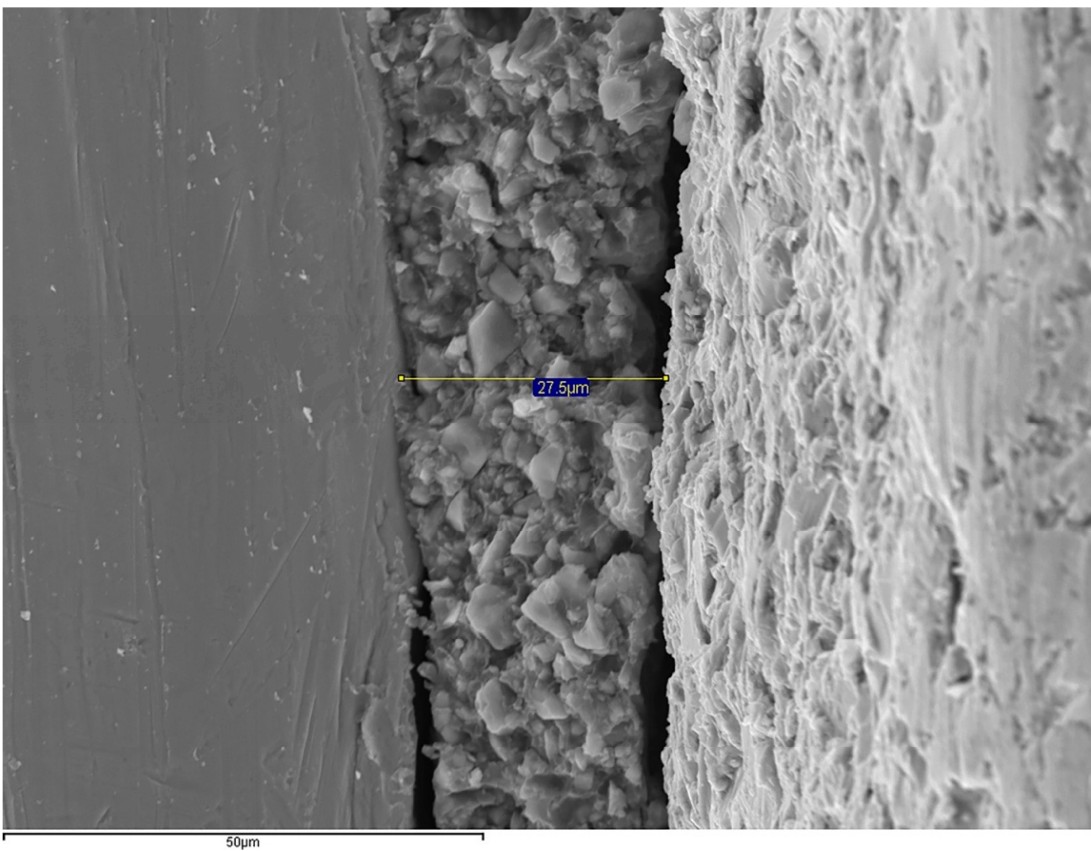

**Figure 4.** SEM image (1000×) calibrated with INCA software showing a marginal discrepancy of a CMc framework.

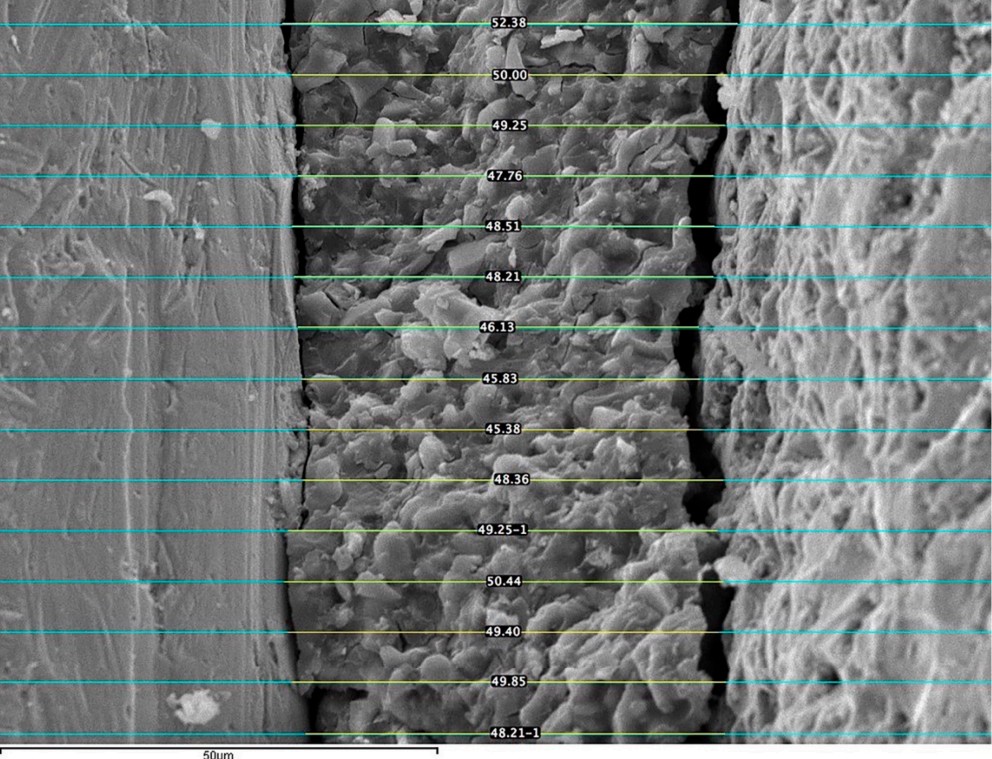

**Figure 5.** SEM image (1000×) showing the marginal discrepancy measurements with Image software of a direct metal laser sintering (LS) framework.

*2.3. Statistical Analysis*

SPSS Version 22.0 (SPSS Inc., Chicago, IL, USA) was used for analyzing the data. The mean values and standard deviations (SD) per group were calculated. The Kruskal–Wallis test and post hoc test for multiple comparisons were used for comparisons among the manufacturing techniques. The Mann–Whitney U test was used to compare both framework designs. The level of statistical significance was set to $\alpha = 0.05$.

## 3. Results

The overall mean and standard deviation marginal discrepancies values of intermediate and cantilever pontic frameworks are listed in Tables 2 and 3, respectively.

**Table 2.** Mean, standard deviation (SD), minimum and maximum (μm) marginal discrepancies values of in frameworks with intermediate pontic.

| Group 1 | Mean | SD | Minimum | Maximum |
|---------|------|------|---------|---------|
| CM | 40.11 | 10.43 | 26.90 | 53.90 |
| LS | 41.84 | 10.36 | 27.16 | 57.20 |
| SM | 39.78 | 8.77 | 24.24 | 53.10 |
| HM | 38.67 | 18.27 | 22.83 | 80.01 |

**Table 3.** Mean, standard deviation (SD), minimum and maximum (μm) marginal discrepancies values in frameworks with cantilever pontic.

| Group 2 | Mean | SD | Minimum | Maximum |
|---------|------|------|---------|---------|
| CMc | 34.55 | 6.93 | 26.94 | 50.71 |
| LSc | 35.22 | 10.60 | 26.17 | 62.42 |
| SMc | 40.81 | 5.55 | 32.18 | 49.21 |
| HMc | 34.17 | 15.85 | 19.63 | 71.69 |

All experimental groups obtained marginal discrepancy values below 50 μm. For the frameworks with an intermediate pontic, the HM group showed the lowest misfit values ($38.67 \pm 18.27$ μm), while the LS subgroup exhibited the highest misfit values ($41.84 \pm 10.36$ μm). The Kruskal–Wallis test revealed no significant differences ($p = 0.677$) in the marginal fit among the different manufacturing techniques. When the cantilever frameworks were analyzed, the HMc group also displayed the lowest marginal discrepancies ($34.17 \pm 15.85$ μm), while the highest misfit values were observed in the SMc group ($40.81 \pm 5.55$ μm). Likewise, no differences ($p = 0.067$) were observed among the manufacturing techniques.

The influence of framework design on the marginal fit was also analyzed, and the Mann–Whitney U test showed no significant differences in any of the tested subgroups: CM and CMc ($p = 0.280$); LS and LSc ($p = 0.123$); SM and SMc ($p = 0.912$); HM and HMc ($p = 0.436$).

## 4. Discussion

This study evaluated the vertical marginal discrepancy of 3-unit Co–Cr posterior FPDs frameworks with intermediate and cantilever pontic and four different manufacturing techniques. The results of the study showed that all tested groups were able to obtain restorations with an adequate marginal fit within the clinically accepted limits of 120 μm [28,29,33,34,43,44], and support the acceptance of the null hypotheses because no significant differences were found among the manufacturing techniques and between the design of the frameworks.

Up to date, there is no consensus on which metallic framework exhibits the best marginal fit. In the study, the CM group on conventional frameworks with intermediate pontic obtained misfit values similar to other studies [18,39] and lower than other ones [17,20,45,46]. The lower values obtained in the study may be due to the fact that the wax patterns of the frameworks were not made with the conventional manual technique, but they were obtained with the 3D-printing technique, in agreement with the study of

Fathi et al. [47] which obtained better fit in cast crowns with wax patterns fabricated by additive CAD/CAM technique. In the last years, several studies have been published about the marginal fit of restorations made with additive technology. In the study, no differences were shown between CM and LS frameworks. While some authors [48–51] have found better fit values in cast structures compared to sintered ones. Conversely, other authors [15,17,18,45,46,52,53] obtained better fit values in selective laser sintering structures. Therefore, there is no consensus regarding the influence of the LS technique on the marginal fit of the restorations. The HM group obtained the lowest misfit values, although there were no differences with the other groups. Afify et al. [54], in Ni-Cr frameworks, and Tamac et al. [55] obtained the same conclusion in their studies comparing cast, sintered and milled crowns. Neese et al. [16] also reported that the milled structures presented a better fit. However, other authors [56,57] reported a better adaptation of the sintered structures. There are only a few studies that compared the four available techniques for manufacturing metal frameworks, and the differences among studies continue. Recent studies [13,58–60] concluded that milled soft metal structures obtain lower misfit values than the other technologies, while other studies [61] continue to advocate for sintered structures. In the study, no differences could be demonstrated among the four manufacturing techniques. Therefore, it remains unclear nowadays which manufacturing technique offers more advantages regarding the marginal adaptation of the metallic restorations.

In certain situations, the disposition of the abutments may not be ideal for FPDs design, and it is necessary to select cantilevered structures as a treatment option [62]. In the present study, the HMc frameworks also obtained the lowest misfit values, although; no differences were demonstrated with the other cantilevered groups. No previous studies were found comparing different manufacturing techniques on cantilever frameworks regarding the marginal adaptation. Therefore, it was not possible to compare the results of the study with previous studies.

In the study, no differences in marginal adaptation were observed when comparing both frameworks design fabricated with the same technology. Likewise, no previous studies were found to compare the results of the study. Previous studies indicated that cantilevered metal–ceramic FPDs can be used effectively in posterior sectors [62], using non-noble alloys, and to replace a tooth with at least two abutment teeth [63,64]. The evidence on the risk of failure of cantilevered FPDs is controversial when compared with the conventional FPDs with an intermediate pontic. Some authors demonstrated that the survival rates of cantilevered prostheses were lower and with more complications [62,64], while others showed a comparable acceptable survival [65–68], although it may vary according to the variables analyzed. Despite the fact that studies on the behavior of cantilevered FPDs are limited, it is important to evaluate this type of prosthesis because it is a treatment option as an alternative to implants or removable partial dentures in daily clinical practice.

The different results among the studies may be due to the different methodology used and the absence of standardization. Several methods have been proposed for measuring the marginal fit, such as silicone replica [51,69,70], direct-view techniques [28,39], profilometry [71,72], or microcomputed tomography [60,73]. In the study, the marginal fit was evaluated by direct viewing with external measurements on an SEM, based on a previous study that demonstrated that destructive methods are not required to assess the marginal fit [28]. There are other aspects that may also directly influence the marginal fit, such as the finish line, the cementation, and the porcelain veneering [26]. In the study, the finish line design was a chamfer, being the most used finish line in recent studies [16,18,20,28,29,50,53,57]. The marginal fit measurements were performed on cemented frameworks to replicate the clinical practice. Previous studies have found higher misfit values after cementation [30,31,43,50], probably due to the hydraulic pressure or the excess of cement [26]. In the study, glass–ionomer cement was used due to its reduced layer thickness [24], and the predetermined internal space for the luting agent was 50 μm in all the frameworks, allowing an adequate cement flow as previously reported [29]. In addition, the luting procedure was carried out in a standardized way, under a seating force

of 50 N, following previous studies that used as reference the 5 Kgf (49 N) [13,24,53,69]. Regarding the veneering porcelain, there is no consensus on whether it affects the marginal fit [18,43,51,74], and in the study, it has not been assessed because the purpose was to analyze only the behavior of the alloy by itself regardless of the veneering ceramic.

Furthermore, the marginal adaptation can also be affected by the manufacturing technique and the Co–Cr alloy selected [60]. Although traditional techniques are handmade processes susceptible to error in any of its phases, and CAD-CAM technologies are automated processes, several factors can affect the marginal fit of the CAD-CAM restorations, such as the accuracy of the scanner, the data scanned transformation to three-dimensional models, and the machine precision [51,75]. In the study, it was analyzed the same Co–Cr alloy with four manufacturing techniques, and it has demonstrated that it provided high precision (below 50 μm) regarding the marginal adaptation, although it may also be due to the precision of the digital technologies employed, including the design and manufacturing of the wax patterns in the casting group.

The study had some limitations. It was performed under standardized conditions, avoiding several variables affecting the clinical practice; however, this allowed testing the marginal adaptation of the frameworks under the same conditions. Another limitation was that only one Co–Cr alloy was analyzed, and it would be interesting to test the technologies analyzed with other alloys. Further research is needed to establish a standardized and reliable method to assess the marginal fit that allows the comparison among the different studies. In addition, more studies are needed on cantilever prostheses since there is a lack of studies to support their clinical behavior. Additional clinical trials are necessary to validate the new technologies in order to achieve greater optimization and standardization. Furthermore, there is a need to review the range of clinical acceptance since digital technologies demonstrate high precision.

## 5. Conclusions

Within the limitations of this in vitro study, the casting technique with the wax patterns obtained by CAD design and 3D printing, and the CAD/CAM technologies as direct metal laser sintering, hard metal milling and soft metal milling seem to guarantee comparable and clinically acceptable misfit values for fabricating Co–Cr posterior FPDs frameworks with intermediate and cantilever pontic.

**Author Contributions:** All the authors contributed to the study, writing, review, and editing of the manuscript. Conceptualization and methodology: C.T., V.R., C.L.-S., J.P., J.C.-B.B. and M.J.S.; supervision: M.J.S. and J.P. Data curation, data visualization: C.T., V.R., C.L.-S., J.P. and M.J.S. writing—reviewing and editing: C.T., V.R., C.L.-S., J.P., J.C.-B.B. and M.J.S. All authors have read and agreed to the published version of the manuscript.

**Funding:** This study was funded by a research grant between the University Complutense of Madrid and Prótesis S.A. (No 381/2015) through the last author.

**Institutional Review Board Statement:** Not Applicable.

**Informed Consent Statement:** Not applicable.

**Data Availability Statement:** The data presented in this study are available on request from the corresponding author.

**Acknowledgments:** The authors would like to thank dental laboratories Prótesis SA., 3Dental, and Dental Creative for manufacturing the frameworks; and Carmen Bravo, Center of Data Processing, Computing Service for Research Support, University Complutense of Madrid, for her assistance with the statistical analysis.

**Conflicts of Interest:** The authors reported no conflict of interest related to this study.

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
