# Peer review of "Effect of Digital Technologies on the Marginal Accuracy of Conventional and Cantilever Co–Cr Posterior-Fixed Partial Dentures Frameworks"

_applsci, doi:10.3390/app11072988_

Round 1

Reviewer 1 Report

Well-written manuscript. Co-Cr fixed partial denture is still in need although implant dentistry has been widely implemented.

There are minor typos seen; please check throughout the manuscript. 

The critical one is the title?  Co-Co Posterior Fixed.... must be Co-Cr?

Reviewer 2 Report

The subject of this paper is novel and is of interest to general dentists as it examines some of the manufacturing options available to clinicians.  Overall, the manuscript is sound, however, it requires a number of clarification and changes:

Major

Grammar and spelling: Some sentences need to be rewritten to more clearly communicate the meaning.

Methodology: Need to give more details on how measurements of marginal misfit were taken, particularly, which sites were examined under SEM and why were those sites chosen as opposed to others?

Minor

Method: Why was 11% shrinkage compensation used? Is it not preset into the CAD software?

References in regards to PFM being the most widely used option for fixed prostheses are at least well over 10 years old, need to either find more recent references or remove sentence altogether.

Reviewer 3 Report

“The aim of this study was to evaluate the effect of different manufacturing techniques, and pontic design on the vertical marginal fit of cobalt-chromium (Co-Cr) posterior fixed partial dentures (FPDs) frameworks”.

General remarks

This study is well conducted by the authors and is recommended for publication.

However, I have some questions that need to be fixed.

1) In the title: “Effect of Digital Technologies on the Marginal Accuracy of Conventional and Cantilever Co-Co Posterior Fixed Partial Dentures Frameworks”

Should be written “(Co-Cr)” instead of “(Co-Co)”

2) In the “Materials and Methods” section,

Experimental Model”

In line 5 is written “… with an intermediate pontic (0.5 mm between abutments)”-according to Figure 1, the distance between the abutments is more than 0.5 mm.

3) Please correct reference No 47- All is written with capital letters
